# Hypercoagulability as Measured by Thrombelastography May Be Associated with the Size of Acute Ischemic Infarct—A Pilot Study

**DOI:** 10.3390/diagnostics11040712

**Published:** 2021-04-15

**Authors:** Adam Wiśniewski, Aleksandra Karczmarska-Wódzka, Joanna Sikora, Przemysław Sobczak, Adam Lemanowicz, Karolina Filipska, Robert Ślusarz

**Affiliations:** 1Department of Neurology, Faculty of Medicine, Nicolaus Copernicus University in Toruń, Collegium Medicum in Bydgoszcz, 85-094 Bydgoszcz, Poland; 2Experimental Biotechnology Research and Teaching Team, Department of Transplantology and General Surgery, Faculty of Medicine, Nicolaus Copernicus University in Toruń, Collegium Medicum in Bydgoszcz, 85-094 Bydgoszcz, Poland; akar@cm.umk.pl (A.K.-W.); joanna.sikora@cm.umk.pl (J.S.); przemyslawsobczak02@gmail.com (P.S.); 3Department of Radiology and Diagnostic Imaging, Faculty of Medicine, Nicolaus Copernicus University in Toruń, Collegium Medicum in Bydgoszcz, 85-094 Bydgoszcz, Poland; adam.lemanowicz@gmail.com; 4Department of Neurological and Neurosurgical Nursing, Faculty of Health Sciences, Nicolaus Copernicus University in Toruń, Collegium Medicum in Bydgoszcz, 85-821 Bydgoszcz, Poland; karolinafilipskakf@gmail.com (K.F.); robert_slu_cmumk@wp.pl (R.Ś.)

**Keywords:** thrombus, stroke, thrombelastography, hypercoagulability, infarct size, ischemic infarct, stroke volume

## Abstract

Background: Thromboelastography (TEG^®^) measures coagulation function in venous blood. Previous studies have reported that this device providing an integrated data on dynamics of clot formation may be useful for predicting clinical outcome in ischemic stroke. We investigated whether a hypercoagulability detected by thrombelastography may be associated with larger size of acute ischemic infarct. Methods: We included 40 ischemic stroke subjects with large artery atherosclerosis or small-vessel disease to a cross-sectional pilot study. Thrombelastography parameters related to time of clot formation (R- reaction time, K-clot kinetics), clot growth and strengthening (angle-alpha and MA-maximum amplitude) and lysis (Ly30) were performed within first 24 h after the onset of stroke. A volume of ischemic infarct was assessed on the basis of diffusion-weighted imaging (DWI) sequence of magnetic resonance imaging. Results: In the entire group, we reported that subjects with a large ischemic focus (>2 cm^3^) had a higher diameter of a clot (measured as MA) than subjects with a small ischemic focus (*p* = 0.0168). In the large artery atherosclerosis subgroup, we showed a significant correlation between MA and size of acute infarct (R = 0.64, *p* = 0.0138), between angle (alpha) and size of acute infarct (R = 0.55, *p* = 0.0428) and stroke subjects with hypercoagulability (MA > 69 mm) had significantly higher probability of a larger size of acute ischemic focus compared to normalcoagulable subjects (5.45 cm^3^ vs. 1.35 cm^3^; *p* = 0.0298). In multivariate logistic regression hypercoagulability was a predictor of a large size of ischemic infarct (Odds ratio OR = 59.5; 95% confidence interval (CI) 1.08–3558.8; *p* = 0.0488). Conclusions: We emphasized that thrombelastography, based on the parameters related to clot strength, may have clinical utility to identify the risk of the extensive ischemic infarct.

## 1. Introduction

Thrombelastography (TEG^®^) measuring the dynamics of clot formation over time and its strength in venous blood may have advantages over the standard tests based on static endpoints (e.g., bleeding time or prothrombin time) and more accurately reflect coagulation processes [1,2]. Hypercoagulability still remains the baseline background of a higher risk of cerebrovascular events, incluing ischemic stroke [3,4]. Many studies have shown that high platelet reactivity, higher levels of fibrinogen or hyperactivated properties of the coagulation system may correspond to the increased incidence of recurrent vascular events [5,6]. Therefore, TEG^®^ is widely applicated in the evaluation of clot transformation and monitoring the coagulation properties of blood in different branches of medicine [7,8,9,10,11]. Recently, increasingly data supported the potential use of this tool in a clinical approach in stroke neurology [12]. Previous studies demonstrated that coagulation parameters recorded in TEG^®^ may be associated with different aspects of stroke management, including early neurological deterioration, prediction of clinical outcome or the risk of hemorrhagic transformation [13,14,15,16,17]. The authors also emphasize its utility in monitoring the safety of intravenous reperfusion therapy [18]. We hypothesize that an individual’s tendency to form firm and large clots in a short period of time may promote the expansion of the ischemic focus in the brain. Therefore, we investigated whether a hypercoagulability that could be detected by TEG^®^ may be associated with the larger size and the extent of acute ischemic lesions in the brain, which undoubtedly contribute to an unfavorable clinical outcome and recovery.

The aim of the current pilot study is to estimate whether the thrombelastography parameters obtained in the acute phase of stroke are related to the extent and volume of the ischemic infarct in the brain.

## 2. Materials and Methods

### 2.1. Study Design and Participants

This cross-sectional, single-center study was conducted from November 2019 to November 2020 in a Stroke Center in the Department of Neurology at the University Hospital No. 1 in Bydgoszcz, Poland. We included 40 subjects who met the clinical and radiological criteria of ischemic stroke in accordance to updated definition proposed by American Heart/Stroke Association. All the stroke subjects were treated with a dose of 150 mg of aspirin. Standard procedures and evaluation methods were performed, including 24 h Holter electrocardiogram monitoring, carotid artery doppler ultrasound, transthoracic echocardiography and transcranial doppler. We enrolled participants with large artery atherosclerosis (atherosclerotic plaque narrowing the lumen of the internal carotid artery at least 50% on the side correspond to stroke symptoms) and small-vessel disease (typical morphological lesions in deep white matter of the brain in the neuroimaging). We excluded stroke subjects with cardioembolic etiology of stroke due to the separate mechanisms of the coagulation cascade, other factors involved in thrombogenesis and related to hypercoagulability, and the use or need to include oral anticoagulants, which would have an impact on the obtained results [19].

The other exclusion criteria were the onset of stroke over 24 h before enrollment, reperfusion therapy (intravenous thrombolysis and/or endovascular treatment), severe strokes which, due to the clinical condition, were incapable of signing a consent to participate, contraindications to magnetic resonance imaging, a history of stroke or myocardial infarction in the previous 3 years, administration of antiplatelet agents or oral anticoagulants before enrollment.

### 2.2. Thrombelastography

The processing device for measurement of coagulation processes was TEG^®^ 5000 Thrombelastograph Hemostasis Analyzer (manufactured by Haemonetics Corp., Braintree, MA, USA). Reactions were carried out in test cells (called cups). The Kaolin TEG^®^ was used for the study. The procedure was performed according to the manufacturer’s instructions. In brief: The whole blood samples were collected from forearm veins within the first 24 h after the onset of stroke. First, we add 1 mL of blood to the reagent container, close it and mix the contents. After we put 20 µL of calcium chloride into the cup, use a pipette to measure 340 µL of the previously prepared solution (blood and kaolin). Then the contents of the cup are set into an oscillating motion. Then a clot also forms. The electrical signal arises from the conversion of vibrations by transformer. The results are presented in a graphic form from which the following values can be read: Reaction time (R)—means the time until the first clot is detected; clots kinetics (K)—the time from the end of the test until the clot reaches 20 mm, which is the rate of clot formation; angle (alpha)—means the rate of fibrin deposition and cross-linking. The angle refers to K and measures the rate of clot growth and strengthening. Maximum Amplitude (MA)—is a reflection of the maximum strength and firmness of the clot; Lysis at 30 min (LY30)—is the percentage of lysed clot after 30 min. We set a limit value of MA over 69 mm as a marker of hypercoagulability, similarly to other studies [16,20]. All subjects with MA values below 69 mm were defined as normalcoagulable.

### 2.3. The Volumetric Evaluation

Magnetic resonance imaging (MRI) was performed within the first 48 h from the onset of stroke symptoms by a 1.5 tesla Optima 450 w scanner (G.E. Healthcare, Chicago, IL, USA). The investigation was conducted in the Department of Radiology at the University Hospital No. 1 in Bydgoszcz. The quantitative analysis of diffusion-weighted imaging (DWI) was assessed by standard diagnostic software package (Functool 4.4, Advantage Workstation 4.4, G.E.Healthcare, Chicago, IL, USA). We set proper threshold values of signal intensities in the series of DWI images to remain only voxels overlapping with the ischemic focus. Then, we manually removed other voxels located outside the area of interest (e.g., noise) within the selected intensity range. Finally, we used a special dedicated function for automatic calculation of the volume of displayed voxels. We provided a final result of the size of acute ischemic infarct in cm^3^. On the basis of obtained volumes, we subdivided the participants into strokes with large and small acute ischemic size, setting the limit value at a volume of 2 cm^3^.

### 2.4. Ethical Statement

Approval of a study protocol by the Bioethics Committee of Nicolaus Copernicus University in Torun at Collegium Medicum of Ludwik Rydygier in Bydgoszcz (KB number 734/2019) was received on 29 October 2019. All participants before enrollment read the study protocol and signed informed consent to have been included to the study. The study was conducted in accordance with the Declaration of Helsinki.

### 2.5. Statistical Evaluation Methods

The statistical calculations were performed with the STATISTICA, version 13.1 (Dell company, Round Rock, TX, USA). The obtained data were expressed as median and range. Non-parametric tests were used for estimation a statistical significance: Mann–Whitney U test (comparison of the continuous values between two groups), Fisher’s exact test (relation between categorized variables), Spearman’s rank correlation test (correlation between the volume of infarct and TEG^®^ values). Univariate and multivariate logistic regression models were used to assess the predictive properties of the obtained thrombelastography parameters. The level of *p* < 0.05 was considered as the threshold for statistical significance.

## 3. Results

### 3.1. The Entire Group

The overall data of the included participants are shown in Table 1.

No significant correlations were found between thrombelastography values (R, K, angle (alpha), MA and Ly30) and the size of acute ischemic infarct in the entire group. We reported only a trend between MA and the volume of stroke (R = 0.36, *p* = 0.0583). Comparison between large and small ischemic infarcts is presented in Table 2.

In the entire group we reported that stroke subjects with large ischemic focus had a significantly higher diameter of a clot (measured as MA) than subjects with small ischemic focus (Figure 1). However, no significant difference in the volume of infarct between hyper- and normalcoagulable subjects was found (2.11 cm^3^ vs. 0.98 cm^3^; *p* = 0.1741).

We performed univariate logistic regression in the entire group to determine predictors of a large size of acute ischemic infarct (Table 3). We reported only a trend in relation to hypercoagulability, whereas the stroke severity (measured by the National Institutes of Health Stroke Scale) was significantly associated with larger size of acute ischemic focus.

Multivariate logistic regression performed in the entire group adjusted for sex, etiology of stroke, hypercoagulability (categorical variables) and age; NIHSS score (continuous variables) showed that hypercoagulability was associated with the higher risk of large ischemic focus (Table 4).

### 3.2. Large Artery Atherosclerosis and Small Vessel Disease

Comparison of subpopulations with large artery atherosclerosis (n = 15) and small vessel disease (n = 25) are presented in Table 5. It is notable, that both subpopulations had similar percentage of hyper- and normalcoagulable subjects and no significant differences were reported between thrombelastography values related to clot growth and strength (MA and angle).

In the large artery atherosclerosis subgroup we showed a significant correlation between MA and size of acute infarct compared to small-vessel disease subgroup (R = 0.64, *p* = 0.0138 vs. R = 0.27, *p* = 0.3529, respectively). Similar dependencies were noted between angle (alpha) and size of acute infarct (R = 0.55, *p* = 0.0428 vs. R = 0.23, *p* = 0.4331, respectively).

Stroke subjects with hypercoagulability (distinguished by MA) in large artery atherosclerosis group had significantly higher probability of a larger size of acute ischemic infarct compared to normalcoagulable subjects (5.45 cm^3^ vs. 1.35 cm^3^; *p* = 0.0298). In a small vessel disease subgroup no significant dependencies were obtained (0.86 cm^3^ vs. 0.58 cm^3^; *p* = 0.9497) (Figure 2).

## 4. Discussion

To our best knowledge we are the first to highlight the association between thrombelastography values and the size of acute ischemic infarct. We emphasized that TEG^®^ parameters reflecting the rate of clot growth and strengthening and especially its firmness may predict a larger volume of ischemic focus. We confirmed the clinical utility and prognostic value of thrombelastography in the acute phase of ischemic stroke.

Previous studies signaled the potential benefits of a comprehensive coagulation assessment for stroke evaluation. Thrombelastography measures the dynamics of clot formation and dissolution and provides more complex and integrated measurement of venous blood coagulation properties than conventional tests [21]. Therefore, thrombelastography may contribute to a more reliable and accurate assessment of hypercoagulability which remains the undoubted pathological cause of ischemic stroke. Elliot et al. [22] showed that stroke subjects are more hypercoagulable than controls, which translates into shorter clot formation times (R and K) and higher parameters proving clot’s strengthening and its total firmness (angle and MA). Several studies have highlighted the potential usefulness of TEG^®^ coagulation monitoring and hypercoagulability detection in predicting clinical outcomes and prognosis in ischemic stroke subjects. Yao et al. [16] showed that elevated maximal clot strength (MA) is related to more severe clinical condition on admission and unfavorable functional outcome at twelve months. Shi et al. [14] revealed that shorter time of clot formation (R) is associated with early neurological deterioration. However, the prognostic value of thrombelastography in predicting the clinical response to intravenous reperfusion therapy has not been confirmed [18]. Nevertheless, this tool may be useful for monitoring the hemostatic processes in stroke subjects undergoing the alteplase therapy [15] and in predicting the risk of hemorrhagic transformation after intravenous thrombolysis [18]. Moreover, the predictive properties of thrombelastography in identifying the subjects with an increased risk of hemorrhagic transformation, regardless of intravenous thrombolysis, have been also underlined [13].

However, a few reports mentioned the potential impact of TEG^®^ parameters on brain lesions. Kawano-Castillo et al. [23] showed that prolonged K values indicated a slower clot formation may be related to hematoma enlargement in stroke subjects with intracranial hemorrhage. The only study that focused on MRI findings among ischemic stroke subjects was performed by Shi et al. The authors emphasized that decreased time of clot formation (shorter R in TEG^®^) is significantly associated with progression in MRI lesions, defined as new, visible changes or enlargement of the initial DWI region, compared with the baseline MRI [14]. However, they analyzed only comparison between two neuroimaging investigations and did not evaluate the volumetric assessment. Therefore, the current study is a preliminary in estimation the relationships between TEG^®^ parameters and the size of ischemic infarcts. Our findings suggest that hemostatic processes related to clot enlargement and strengthening more than the initial dynamics of clot formation are involved in the risk of extension of ischemic lesions in acute phase of stroke. The increased maximum clot strength (MA in TEG^®^), reflecting hypercoagulability, became the most significant value on thrombelastography as it remained more frequent in subjects with larger ischemic infarct volume compared to small infarct volumes, and above all, an independent predictor of increased ischemic infarct size. Moreover, the obtained relationships applied to the entire study group and additionally reached higher statistical significance, especially in the subgroup with large artery atherosclerosis etiology of stroke. Furthermore, in this subgroup, we indicated that also higher values of angle (alpha) promote a larger size of ischemic infarct. Thus, we emphasized that thrombelastography may be particularly useful in predicting the size of ischemic focus in subjects with carotid artery pathology as a cause of stroke. We hypothesize that above dependencies may result from coexisting hyperreactivity of platelets, which is especially strongly expressed and important in the pathogenesis of large artery atherosclerosis compared to other stroke etiologies [24]. The role of platelet activity in this subgroup appears to be primary and indisputabled, whereas in small-vessel disease other processes are predominant, such as calcification, fibrosis or glazing [25]. We reported also in our previous study that high on-treatment platelet reactivity promotes larger size of ischemic focus only among large artery atherosclerosis subjects [26]. Huang et al. [27] reported that thrombelastography parameters indicating clot strength (angle and MA) significantly correlate with higher platelet reactivity. Moreover, Yao et al. showed that subjects with large artery atherosclerosis stroke subtype are more likely to achieve high levels of MA in TEG^®^ [16]. Furthermore, TEG^®^ values are associated with the platelet composition of clots. Elliot et al. [22] showed that elevated maximal clot strength indicates the formed clot with stronger platelet-fibrin matrices, and Barua et al. [28] reported that 87% of the total maximum amplitude of clot is attributable to platelet–fibrin interaction. The above considerations may account for the particular value of our findings in relation to the subgroup with large artery atherosclerosis. We did not reveal any significant impact of TEG^®^ parameters measuring the time of clot formation (R and K) on the size of ischemic infarct. We hypothesize that it could be due to the platelets being less involved in these stages of clot formation. The time to formation of the basal clot mass may have a greater influence on the course of stroke, as other authors have shown, not affecting the size of ischemic lesion.

However, two important issues should be also highlighted. First, there is a lack of thrombelastography data in healthy, sex and age-matched control group. We analyzed only TEG^®^ parameters among stroke subjects and compared groups with two different etiologies of stroke. Second, it is possible that observed associations may be due to reverse causation. Therefore, in our study we avoid comments about the influence of thrombelastography on the size of the infarcts and emphasize the existence of interdependencies.

We are fully aware of the limitations. We have conducted this research as a pilot study, based on a small sample size. Therefore, our findings should be viewed with caution and need to be confirmed and verified on larger cohorts. Our analysis did not include all strokes, especially subjects with extensive ischemic areas, in severe clinical condition, due to the inability to obtain informed consent. The authors also take the position that the measurement of venous blood coagulation by thrombelastography does not have to correspond and accurately reflect the characteristics of the coagulation processes in the cerebral arterial circulation.

## 5. Conclusions

In summary, in this pilot study we emphasized that hypercoagulability, based on thrombelastography parameters related to clot strength, is associated with larger size of acute ischemic infarct. Particularly in large artery atherosclerosis stroke subtypes it may be beneficial to monitor the coagulation properties to identify and stratify the risk of extensive ischemic areas in the brain. Therefore, our novel findings contribute to the expand the existing knowledge on predictive values of TEG and support its clinical usefulness in the acute phase of ischemic stroke. Nonetheless, due to the preliminary nature of this research, further studies are necessary.

## Figures and Tables

**Figure 1 diagnostics-11-00712-f001:**
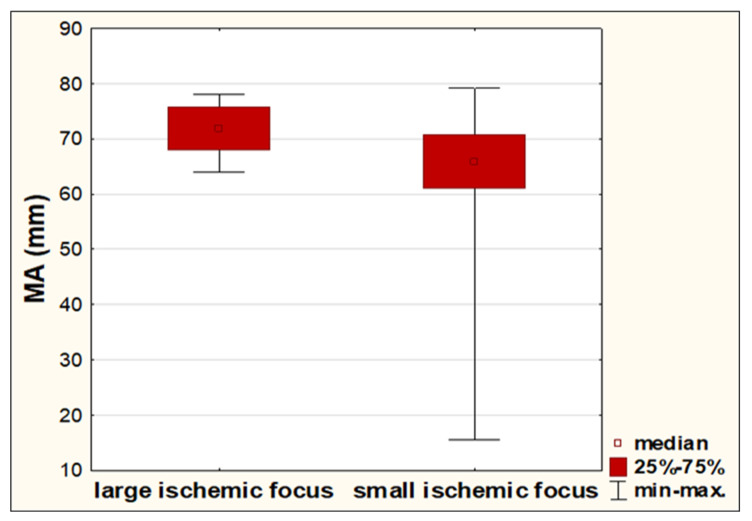
Differences in diameter of clot (measured as MA-maximum amplitude in mm) between stroke subjects with large (>2 cm^3^) and small (<2 cm^3^) ischemic infarcts.

**Figure 2 diagnostics-11-00712-f002:**
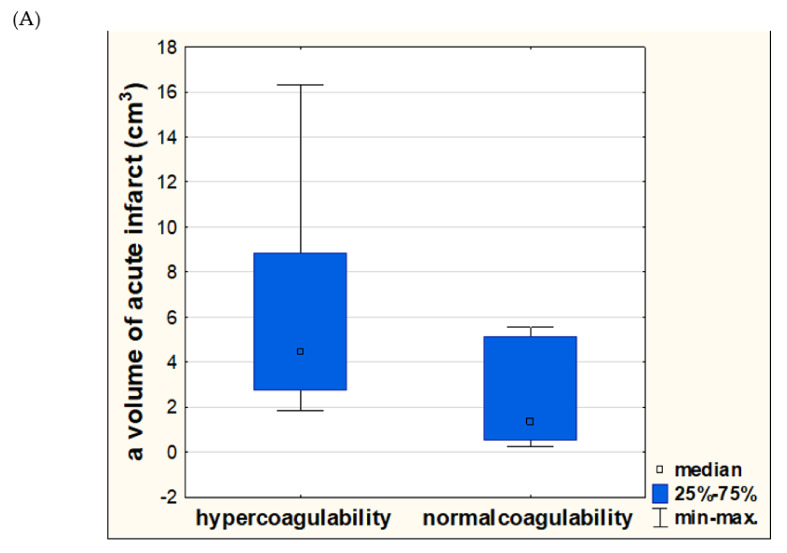
Differences in a volume of acute ischemic infarct (measured in cm^3^) between stroke subjects with hyper- or normalcoagulability, distinguished by maximal amplitude in thrombelastography. Significant difference was reported only in a large artery atherosclerosis subgroup (**A**). No significant differences were noted in a small-vessel subgroup (**B**).

**Table 1 diagnostics-11-00712-t001:** The baseline characteristics of ischemic stroke subjects (n = 40).

Parameter	Value
Age in years, median (range)	68.5 (43–89)
Sex, male, N (%)	21(52.5%)
Large artery atherosclerosis, N (%) Small vessel disease, N (%)	15 (37.5%) 25 (62.5%)
Hypercoagulability, N (%)	21 (52.5%)
NIHSS (points) at admission, median (range) NIHSS (points) at discharge, median (range)	4 (2–15) 3 (1–14)
mRS (points) at admission, median (range) mRS (points) at discharge, median (range)	2 (1–5) 1 (0–5)
R (minutes), median (range) K (minutes), median (range) Ly30 (%), median (range) MA (mm), median (range) Angle (alpha), degrees, median (range)	8.85 (2.4–21.3) 2.3 (1.1–5.8) 0.25 (0–78.1) 69.25 (15.6–79.8) 55.6 (24.3–73.1)
Volume of acute infarct (cm^3^), median (range)	1.46 (0.09–136.45)
Large infarct, (>2 cm^3^), N (%) Small infarct (<2 cm^3^), N (%)	14 (35%) 26 (65%)
Hypertension, N (%)	29 (72.5%)
Diabetes, N (%)	12 (30%)
Hyperlipidemia, N (%)	13 (32.5%)
Smoking, N (%)	19 (47.5%)
Obesity, N (%)	8 (20%)
CRP (mg/L) median (range)	2.37 (0.21–67.4)
HBA1c (%) median (range)	5.8 (4.8–13.1)
Platelets (thousands/µL) median (range)	268 (102–618)
Fibrinogen (mg/dL) median (range)	363 (110–646)

NIHSS, National Institutes of Health Stroke Scale; mRS, modified Rankin Scale; CRP, C-reactive protein; HBA1c, hemoglobin A1c; R, reaction time; K, clot kinetics; Ly30, percentage of lysed clot at 30 min; MA, maximum amplitude.

**Table 2 diagnostics-11-00712-t002:** Comparison of subpopulations with large (>2 cm^3^) and small (<2 cm^3^) acute ischemic infarct.

Parameter	Large Infarct (n = 14)	Small Infarct (n = 26)	*p*-Values
Age, median (range)	71.5 (56–85)	66.5 (43–89)	0.3268
Sex, male, N (%)	7 (50%)	14 (50%)	0.9145
Hypertension, N (%)	12 (80%)	17 (68%)	0.6114
Diabetes, N (%)	4 (28.6%)	8 (32%)	0.6350
Hyperlipidemia, N (%)	7 (50%)	6 (23%)	0.0343
Smoking, N (%)	8 (57.1%)	11 (42.3%)	0.2632
Obesity, N (%)	2 (14.2%)	6 (23%)	0.3128
CRP (mg/L) median (range)	2.60 (0.97–67.4)	1.72 (0.21–13.4)	0.6259
HBA1c (%) median (range)	5.8 (5.3–13.1)	5.6 (4.8–7.2)	0.1756
Platelet count (thousands/µL) median (range)	285 (159–618)	240 (102–395)	0.2752
Fibrinogen (mg/dL) median (range)	382 (110–646)	363 (241–480)	0.3529
Hypercoagulability, N (%)	9 (64.3%)	12 (46.2%)	0.2233
NIHSS at admission median (range)	8 (2–15)	4 (2–7)	0.0054
NIHSS at discharge median (range)	6 (2–14)	3 (1–6)	0.0043
mRS at admission median (range)	4 (1–5)	2 (0–4)	0.0633
mRS at discharge median (range)	3 (1–5)	1 (0–3)	0.0613
R (minutes), median (range)	9.6 (5.4–21.3)	9 (2.4–18.2)	0.8345
K (minutes), median (range)	2.4 (1.2–5.8)	2.5 (1.1–4.2)	0.5419
Ly30 (%), median (range)	0.2 (0.0–1.0)	0.4 (0.01–78.1)	0.2186
MA (mm), median (range)	71.8 (63.9–78.0)	65.9(15.6–79.8)	0.0168
Angle (alpha), degrees, median (range)	59.2 (43.6–71.7)	54.2 (24.3–73.1)	0.1144
Volume of acute infarct (cm^3^), median (range)	5.5 (2.06–136.5)	0.73 (0.09–1.86)	<0.0001

NIHSS, National Institutes of Health Stroke Scale; mRS, modified Rankin Scale; CRP, C-reactive protein; HBA1c, hemoglobin A1c; R, reaction time; K, clot kinetics; Ly30, percentage of lysed clot at 30 min; MA, maximum amplitude. *p*-Values are estimated based on exact Fisher’s test (categorical variables) or Mann-Whitney U test (continuous variables).

**Table 3 diagnostics-11-00712-t003:** Univariate logistic regression analysis of predictors of a large size (>2 cm^3^) of acute ischemic infarct.

Parameter	OR (95% CI)	*p*
Hypercoagulability	4.4 (0.89, 21.78)	0.0694
Sex (female)	0.92 (0.20, 4.17)	0.9122
Large artery atherosclerosis	3.33 (0.69, 16.02)	0.1328
Age	1.06 (0.98, 1.15)	0.1584
NIHSS score	1.83 (1.15, 2.90)	0.0106 *
Platelet count	1.01 (0.99, 1.02)	0.1531
Fibrinogen	1.00 (0.99, 1.01)	0.4317
Hypertension	1.67 (0.25, 11.07)	0.5970
Smoking	0.71 (0.16, 3.23)	0.6621
Hyperlipidemia	3.0 (0.61, 14.86)	0.1785
Diabetes	5.0 (0.77, 32.57)	0.0923

*—significant dependencies, OR—odds ratio, CI—confidence interval, NIHSS—the National Institutes of Health Stroke Scale.

**Table 4 diagnostics-11-00712-t004:** Multivariate logistic regression model of independent predictors of a large size of acute ischemic focus (>2 cm^3^).

Variables	Adjusted OR (95% CI)	*p*
Sex (female)	0.06 (0.00, 3.71)	0.1823
Age	1.16 (0.96, 1.39)	0.1272
Hypercoagulability	59.05 (1.08, 3558.8)	0.0488 *
Large artery atherosclerosis	19.27 (0.40, 919.2)	0.1335
NIHSS score	1.82 (0.90, 3.69)	0.0976

*—significant dependencies, OR—odds ratio; CI—confidence interval; NIHSS—the National Institutes of Health Stroke Scale.

**Table 5 diagnostics-11-00712-t005:** Comparison of subpopulations with two etiologies of stroke.

Parameter	Large Artery Atheroscleros (n = 15)	Small Vessel Disease (n = 25)	*p*-Values
Age, median (range)	67 (43–89)	69 (54–85)	0.4418
Sex, male, N (%)	7 (46.7%)	14 (56%)	0.4059
Hypertension, N (%)	13 (86.7%)	16 (64%)	0.4356
Diabetes, N (%)	4 (26.6%)	8 (32%)	0.5050
Hyperlipidemia, N (%)	8 (53.3%)	5 (20%)	0.0343
Smoking, N (%)	9 (60%)	10 (40%)	0.2876
Obesity, N (%)	2 (13.3%)	6 (24%)	0.3545
CRP (mg/L) median (range)	2.68 (0.97–67.4)	1.94 (0.21–18.1)	0.3713
HBA1c (%) median (range)	5.8 (5.1–13.1)	5.8 (4.8–12.4)	0.8885
Platelet count (thousands/µL), median (range)	287 (159–618)	250 (102–353)	0.2697
Fibrinogen (mg/dL) median (range)	361 (110–451)	365 (180– 646)	0.7268
Hypercoagulability, N (%)	8 (53.3%)	13 (52%)	0.5971
NIHSS at admission median (range)	6 (2–15)	3 (2–8)	0.0189
NIHSS at discharge median (range)	5 (1–14)	3 (1–8)	0.0484
mRS at admission median (range)	3 (1–5)	2 (1–5)	0.1964
mRS at discharge median (range)	2 (1–5)	1 (0–5)	0.6700
R (minutes), median (range)	10 (5.4–20.2)	8 (2.4–21.3)	0.0163
K (minutes), median (range)	2.7 (1.5–4.5)	2.3 (1.1–5.8)	0.2076
Ly30 (%), median (range)	0.5 (0.01–78.1)	0.10 (0.00–53.4)	0.1882
MA (mm), median (range)	69.1 (15.6–77.3)	69.4 (58.6–79.8)	0.7799
Angle (alpha), degrees, median (range)	50 (24.3–68.3)	56.5 (36.1–73.1)	0.0992
Volume of acute infarct (cm^3^), median (range)	3.13 (0.23–136.5)	0.87 (0.09–2.65)	0.0291
Large infarct (>2 cm^3^), N (%)	13 (86.7%)	1 (4%)	<0.0001

NIHSS, National Institutes of Health Stroke Scale; mRS, modified Rankin Scale; CRP, C-reactive protein; HBA1c, hemoglobin A1c; R, reaction time; K, clot kinetics; Ly30, percentage of lysed clot at 30 min; MA, maximum amplitude. *p*-Values are estimated based on exact Fisher’s test (categorical variables) or Mann-Whitney U test (continuous variables).

## Data Availability

The data that support the findings of this study are available from the corresponding author upon request.

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
