# Peer review of "Hypercoagulability as Measured by Thrombelastography May Be Associated with the Size of Acute Ischemic Infarct—A Pilot Study"

_diagnostics, 2021, doi:10.3390/diagnostics11040712_

Round 1
Reviewer 1 Report
I have no further comments.
Author Response
Thank You for Your positive comment on our paper.
Reviewer 2 Report
In this manuscript the authors present data about a small pilot study performed on stroke patients and TEG. Although of a possible interest, the presentation of the data must be refined as there could be serious flaws based on the statistical analysis. Also as stated by the authors, the patients already had the diagnosis of ischemic stroke and therefore the study is rather cross-sectional and not prospective. What was the follow-up period as it is not presented?
A table comparing the outcome of the logistic regression should be additionally presented (comparisons between small stroke patients and large stroke patients). How did the authors consider to introduce the variables in the logistic regression, maybe some of them are not associated with the outcome? This data should be presented.
The results reported in the result section and abstract seem to be conflicting. In univariate analysis hypercoagulabilty was not associated with the stroke size but the authors state in the abstract that “in multivariate logistic regression hypercoagulability was an independent predictor of an increased size of ischemic infarct (Odds ratio OR=59.523.55; 95% confidence interval (CI) 0.981.07-3558.8 520.4; p=0.048856 “.However, the 95% CI of the OR includes value 1 and such CI is not statistically significant, the p value should actually be >0.05 (close to statistical significance but still not significant).
In order for the authors to conclude that hypercoagulability is an independent factor for stroke size a linear regression should be performed with the outcome variable the size of stroke. Therefore the conclusion of the article is not based upon the presented data in the manuscript.
Also there is a concern about the fact that the authors performed logistic regression with a relatively high number of independent variables on a small sample size. This aspect associated with the fact that some data are in contradiction suggest that the authors should perform again all the statistical analysis and try to reduce the number of covariates in the model.
Data about stroke etiology is not presented although this variable is reported in the multivariate analysis (do the authors refer to large artery atherosclerosis and small vessel disease?). Otherwise the reason comparing large artery atherosclerosis and small vessel disease is not clear.
As an overall comment the manuscript is very difficult to read as it seems to be a draft. Some figures are obviously deleted but still inserted in the manuscript, making the results section difficult to follow. There seems to be two tables that are numbered with no 2, and most figures only repeat the data that is presented in text.
Data in table 4 suggests that hypercoagulability was an independent factor only on large artery atherosclerosis disease subgroup. As the authors perform subgroup analysis it is not clear if the authors decided to perform post-hoc analysis. Also Fischer test seem to be more appropriate for categorical variables as the sample size is small.
Figure 1 and figure 2 could be deleted
Table 1: you should only present the number of patients with hypercoagulability, and in general only one category of a binary variable.
Round 2
Reviewer 2 Report
Thank you. The authors improved the manuscript enough.
This manuscript is a resubmission of an earlier submission. The following is a list of the peer review reports and author responses from that submission.
Round 1
Reviewer 1 Report
This is a study on the association of the infarct size of an acute ischemic stroke and in vitro measurements of hypercoagulability. The authors analysed 40 patients with stroke after applying numerous exclusion criteria. Coagulation was assessed by thromboelastography and infarct size was determined by MRI. Based on radiological findings, the patients were grouped into those with "large artery arteriosclerosis" and those with "small vessel disease". The authors conclude that thromboelastographic findings of hypercoagulability may help to identify patients with large infarct size.
Major comments: Aim and hypothesis are quite vague ("usefulness of thromboelastography"; p 2, l 61). In parallel, exposure and outcome are not strictly defined. Instead, the authors analyse numerous thromboelastographic findings in the entire study population and in four subgroups (large vs. small vessel disease, large vs small infarct size). This approach is prone to statistical overfitting and the findings may correspond to chance and not to biologically meaningful phenomena. The authors assume that hypercoagulability may cause larger infarct size; however, the association - if true - could be interpreted vice versa.
Minor comments: The authors give baseline characteristics of the 40 patients included in the study (Table 1). However, no information (not even the number of patients) is given for the subgroups. How many patients had a "large" infarct? Is the finding "hypercoagulability" used in the logistic regression dichotomised? The lack of this information makes the interpretation of the data and mainly the logistic regression difficult.
Reviewer 2 Report
It is an interesting paper, however, some suggestions must be critically addressed:
- What about the NIHSS score - my question would be if thromboelastography (hypercoagulability) parameters would correlate with stroke severity as assessed by the NIHSS scale. I think this would be worthy of additional investigation.
- What was the presence of atrial fibrillation in this cohort? This should be discussed.
- It is unclear how these patients were managed? Pharmacological thrombolysis/fibrinolysis, mechanical thrombectomy? This is critical to understand and interpret these results. This must be addressed and critically discussed. You mention that patients receiving reperfusion treatment were excluded. Then with what did you treat these patients, what was their clinical management given that they all had a stroke?
- Please revise the figures presented - use conventional colors and improve/increase the font size of the numbers since this cannot be seen easily. Improve quality/resolution of images provided.
- The regression model should be defined - we do not know explicitly which variables were model adjusted (multivariable regression). This should be, at minimum, adjusted for sex, age, NIHSS score, etc. Please report new results. Also, the confidence interval for hypercoagulability as an independent predictor of large infarct should be explained (1.07 - 520.4) with borderline significance. This is problematic and more stringent statistical methods should be applied with a well-defined regression model.
Round 2
Reviewer 1 Report
Thank you for revising the manuscript. The authors made some changes and improved the hypothesis and the definition of "hypercoagulation". However, important information on the study population and the subpopulations are still missing. Without that information the robustness of the statistical model cannot be estimated.
Reviewer 2 Report
The authors have answered all my queries successfully.
Author Response
Thank You for positive comments regarding our manuscript.
Round 3
Reviewer 1 Report
Thank you for providing the number of patients in each group (Table 2). If I get it right, there are 15 patients with “large artery sclerosis” of whom 13 had a larger infarct, and 25 patients with “small vessel disease”, of whom 1 had a large infarct. Despite the very small number of patients with small infarcts and “large vessel disease” (n=2) and large infarcts and “small vessel disease” (n=1) the authors make comparisons between these groups (Figure 3 B and C; Figure 4 B and C; Table 2 (Univariate analysis…)) and draw conclusions (e.g., Abstract line 29). I do not feel that this approach is appropriate. I strongly suggest that the authors focus on two groups. It is up to them to decide if the comparison between small and large vessel disease is of higher biological importance or the comparison between large and small infarcts. How many patients fulfil the criterion “hypercoagulability”?